# Performance of Insoluble IrO$_2$ Anode for Sewage Sludge Cake Electrodehydration Application with Respect to Operation Conditions

Nam-Young Lee [1], Mi-Young You [2], Jaemyung Lee [3], Seohan Kim [2,4,*] and Pung Keun Song [1,*]

[1] Department of Materials Science and Engineering, Pusan National University, Busan 46241, Korea; nyleee@pusan.ac.kr
[2] The Institute of Materials Technology, Pusan National University, Busan 46241, Korea; miyoung5113@pusan.ac.kr
[3] Hydrogen Ship Technology Center, Pusan National University, Busan 46241, Korea; jmlee1@pusan.ac.kr
[4] Department of Materials Science and Engineering, Ångström Laboratory, Uppsala University, 75321 Uppsala, Sweden
[*] Correspondence: seohan.kim@angstrom.uu.se (S.K.); pksong@pusan.ac.kr (P.K.S.)

**Abstract:** The efficient management of wastewater and sewage sludge treatment are becoming crucial with industrialization and increasing anthropological effects. Dehydration of sewage sludge cakes (SSCs) is typically carried out using mechanical and electrochemical processes. Using the mechanical dehydration process, only a limited amount of water can be removed, and the resultant SSCs have a water content of approximately 70–80 wt.%, which is significantly high for land dumping or recycling as solid fuel. Dumping high-moisture-content SSCs in land can lead to leakage of hazardous wastewater into the ground and cause economic loss. Therefore, dehydration of SSCs is crucial. Contemporary treatment methods focus on the development of anode materials for the electrochemical processes. IrO$_2$ is an insoluble anode material that is eco-friendly, less expensive, and exhibits high chemical stability, and it has been widely used and investigated in wastewater treatment and electrodehydration (ED) industries. Herein, we evaluated the performance of the ED system developed using IrO$_2$ anode material. The operating conditions of the anode such as reaction time, sludge thickness, and voltage on SSC were optimized. The performance of the ED system was evaluated based on the moisture content of SSCs after dehydration. The moisture content decreased proportionally with the reaction time, sludge thickness, and voltage. The moisture content of 40 wt.% was determined as the optimum quantity for land dumping or to be used as recycled solid fuel.

**Keywords:** electrodehydration; insoluble anode material; sewage sludge cake; iridium oxide

## 1. Introduction

With the rapid development of industry and human civilization, wastewater and sewage sludge generation have also increased rapidly [1–3]. Therefore, there is increased focus on methods for treating waste [4–8]. Until now, sewage sludge has been dumped in the ocean, which has currently become impossible owing to the international restriction on ocean and land dumping, as dumping in the ocean results in the release of hazardous chemicals and organics and pollutes the environment [5,7–10]. Processes involved in land dumping such as incineration, dumping, and composting have various disadvantages such as the amount of energy for incineration, requirement of dedicated land, and leachate [9]. Previous studies focused on overcoming these problems by focusing on moisture content. The moisture content should be <40 wt.% for land dumping to prevent leachate and the cost of dumping [9,11,12]. The moisture content should be <50 wt.% for self-thermal combustion.

Dehydration of sewage sludge cake (SSC) has been studied based on mechanical, thermal, and electrodehydration (ED) methods [7,8,13,14]. Water in SSCs can be divided into four categories: (i) free water, (ii) surface water, (iii) interstitial water, and (iv) bound water [15,16]. The most common mechanical dehydration method results in a moisture

content of 70–80 wt.%, which indicates that only free water that is mechanically attached to the surface is removed. It is difficult to remove surface, interstitial, and bound water using mechanical dehydration. Using thermal dehydration, not only the surface water attached to the surface but also the interstitial and bound water inside the sludge pore (floc) can be removed. However, the thermal dehydration method requires an enormous amount of energy. The minimum amount of energy required for water vaporization is approximately 0.617 kwh/kg $H_2O$ [15]. The ED method, however, can extract all types of water in the sludge. Compared to thermal dehydration, the ED method is also attractive from an economic point of view. The ED method can efficiently remove the moisture that is physically and chemically combined with the sludge via electrophoresis and electro-osmosis, because of which it does not require large amounts of energy; in the thermal dehydration method, the water vapor enthalpy technique is used. In general, SSCs are mechanically dehydrated before ED processing; therefore, they do not contain any free water [17,18]. The ED process takes place at various stages. Initially, the solid particles in SSCs move toward the anode under the presence of an electric field. In this stage, residual free water is removed. Subsequently, interstitial and some of the surface water moves toward the cathode via electrophoresis, and some of the bound water moves by capillary force. In this stage, the flocs inside the SSCs are destroyed, and subsequently, the rest of the bound water is removed.

The anode used in the ED system should be resistant to properties such as oxidation, chemicals, corrosion, and thermal degradation. $IrO_2$, boron-doped diamond (BDD), and Pt are considered representative materials [19–21]. However, BDD and Pt are considered expensive for application as anode materials for ED systems. Further research on reducing the fabrication cost is in progress [19,22]. $IrO_2$ electrodes are cost-effective and also exhibit comparable performance to BDD. $IrO_2$ is typically prepared using chemical bath deposition [23,24]. Numerous studies have been carried out on the fabrication methods and electrode characteristics of $IrO_2$ [15,16,20,25]. However, studies on the operation parameters of ED systems with $IrO_2$ anodes are limited. ED systems are widely used in various environmental conditions that exhibit diverse characteristics; therefore, the significance of studying the operating conditions is increased. The key parameters are the thickness of the SSC, voltage, and processing time. The dehydration performance can be increased by increasing the voltage and processing time and by decreasing the thickness of the SSC. However, increasing the voltage is economically disadvantageous owing to the high use of electrical energy. Further, on application of high voltage, there is a possibility that the dehydrated SSC is burned, and it cannot be used as recycled solid fuel. In addition, the application of high voltage can reduce the lifetime of the anode owing to corrosion and delamination.

In this study, we investigated various ED operation conditions such as the thickness of SSC, voltage, and processing time to determine which factor has the most influence on dehydration performance. Optimized conditions that result in high dehydration performance were suggested. The stability of $IrO_2$ was evaluated under various operating conditions, and the performance of the ED system was estimated by comparing the moisture content before and after the ED process. The moisture contents were proportional to the thickness of the SSC, and inversely proportional to the voltage and processing time. The lowest moisture contents obtained was 33.4 wt.% at the following operation conditions: applied voltage of 70 mV/mm$^2$, SSC thickness of 0.5 mm, and processing time of 140 s. However, after ED processing, the $IrO_2$ electrode exhibited cracks on the surface. Therefore, the optimized conditions considering anode lifetime and cost efficiency were determined and are as follows: 70 mV/mm$^2$ voltage, SSC thickness of 1 mm, and processing time of 140 s.

## 2. Materials and Methods

The SSCs used in this study were collected from a sewage treatment plant and a city in the Republic of Korea. The SSCs were dehydrated to 80 wt.% using a mechanical dehydration process and then stored at 2 °C. A laboratory-made, batch type ED system was

used, and the photographs of the ED system are shown in Figure 1. A 50 × 50 mm$^2$ IrO$_2$ anode of 8 µm thickness coated on Ti (thickness = 2 mm) was purchased from Hankyung TNC corporation. SUS304 stainless steel was used as the cathode material. Details of the experimental parameters such as SSC thickness, voltage, and processing time are listed in Table 1.

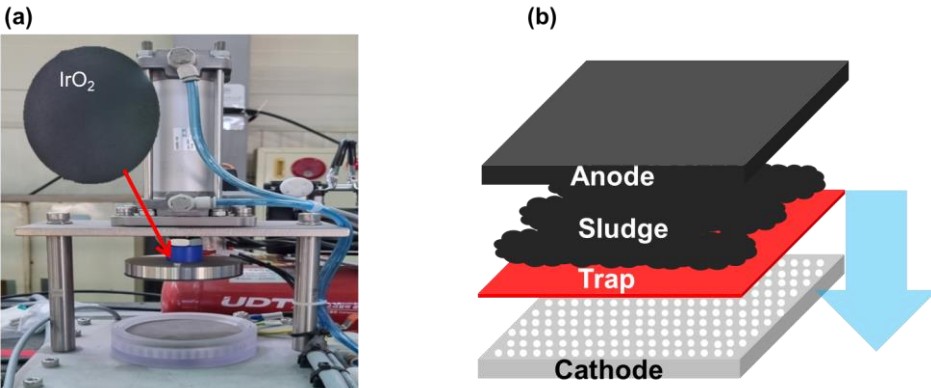

**Figure 1.** (**a**) Photographs of laboratory-made batch type dehydration system; (**b**) schematic image of the anode and cathode assembly. The anode and cathode material are IrO$_2$ and SUS304, respectively.

**Table 1.** Experimental parameters for electrodehydration process using IrO$_2$ anode.

| Experimental Parameter | Values |
|---|---|
| Initial moisture contents (wt.%) | 80 |
| Processing time (s) | 0, 30, 60, 90, 120, 140 |
| SSC thickness (mm) | 0.5, 1, 2 |
| Voltage (mV/mm$^2$) | 10, 30, 50, 70 |

The moisture content was analyzed conforming to waste process test standards of South Korea National Institute of Environmental Research Notice No. 2017-54. The electrochemical characteristics were evaluated using cyclic voltammetry (ZIVE LAB, ZIVE MP2). Pt was used as the counter electrode, and Ag/AgCl was used as the reference electrode. The measurement was conducted in 0.1 M H$_2$SO$_4$ electrolyte solution. The microstructure was observed using X-ray diffraction (XRD, Rigaku, ULTIMA4), and the surface morphology and micro floc were observed using field emission scanning electron microscopy (FE-SEM, TESCAN, MIRA3).

## 3. Results

Figure 2 shows the moisture contents of SSCs after dehydration (using batch type dehydration system (Figure 1)) with respect to voltage and at various SSC thicknesses, such as (a) 2 mm, (b) 1 mm, and (c) 0.5 mm. As previously mentioned, the initial moisture contents of SSCs were 80 wt.%, after mechanical dehydration. The moisture content decreased with an increase in processing time and voltage in all three SSCs with varying thicknesses, as shown in Figure 2a–c. The moisture contents also decreased with a decrease in SSC thickness, and the lowest moisture content (33.4 wt.%) was obtained for SSC with 0.5 mm thickness (Figure 2c). In the case of SSC with 2 mm thickness (Figure 2a), at 10 and 30 mV/mm$^2$ applied voltage, the moisture content did not decrease to 60 wt.%. This indicates that the 2 mm thick SSC requires applied voltage >30 mV/mm$^2$ to completely remove the interstitial and surface water. As shown in Figure 2b,c, 10 mV/mm$^2$ is not an adequate voltage to completely remove the water content. However, an applied voltage of 50 and 70 mV/mm$^2$ for the 2 mm thick SSC resulted in the removal of interstitial and surface water. In addition, for an applied voltage of 70 mV/mm$^2$, the bound water was totally removed in 60 s from 0.5-and 1 mm thick SSCs. In the case of the 0.5 mm thick SSC (Figure 2c), applied voltages of 30, 50, and 70 mV/mm$^2$ resulted in the removal of

interstitial and surface water in 30 s. As mentioned previously, the moisture content should be <40 wt.% for land dumping or to be used as solid recovered fuel (SRF). Applied voltages of 10 and 30 mV/mm$^2$ did not decrease moisture content to <40 wt.%. In the case of 0.5 mm thick SSC, applied voltage of 50 mV/mm$^2$ resulted in moisture content of <40 wt.% in 140 s. In the case of 1 mm thick SSC, an applied voltage of 70 mV/mm$^2$ resulted in moisture content of <40 wt.% in 140 s. At the same applied voltage, the 0.5 mm thick SSC resulted in 33.4 wt.% moisture content in 140 s. This indicates that applied voltages of 50 and 70 mV/mm$^2$ result in the removal of not only the interstitial and surface water but also the bound water and flocs in SSCs.

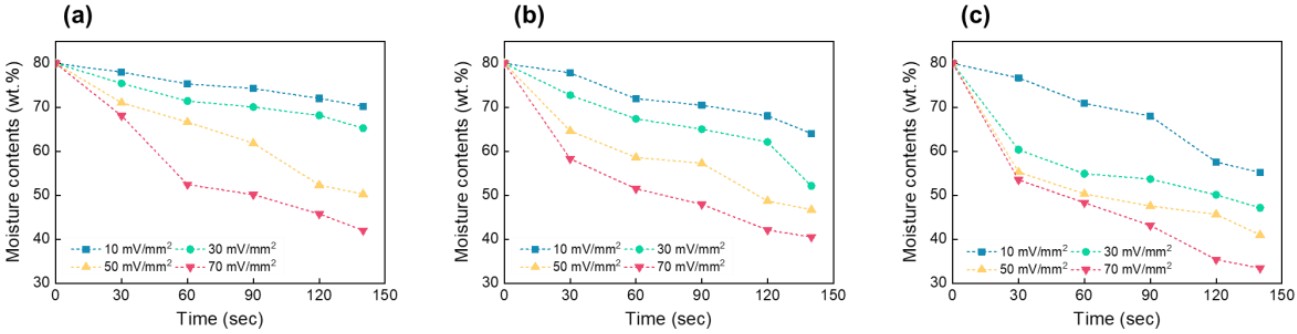

**Figure 2.** Moisture content with respect to processing time of sewage sludge cake thickness (**a**) 2 mm, (**b**) 1 mm, and (**c**) 0.5 mm.

Figure 3 shows dehydration rates of SSC at various thicknesses with respect to voltage and processing time. The dehydration rate was calculated by determining the moisture content before and after the ED process. In general, the ED process occurs in two steps. The first step is the removal of the interstitial and surface water, and the second step is the removal of bound water, which occurs when the flocs present inside the SCCs are destroyed. Applied voltages of 10 and 30 mV/mm$^2$ do not yield a remarkable dehydration rate. An applied voltage of 10 mV/mm$^2$ for SSC with thicknesses >1 mm does not result in any evident dehydration process. In the case of the 0.5 mm thick SSC, an applied voltage of 10 mV/mm$^2$ resulted in the removal of the interstitial, surface, and some of the bound water in 120 s. When the applied voltage was 50 mV/mm$^2$, an evident first peak and a small second peak were observed. It is assumed that the interstitial and the surface water were completely removed, but bound water was still present in the flocs in the SSC. For an applied voltage of 70 mV/mm$^2$, a single peak was observed in all the SSCs of varying thicknesses, showing that all the three types of water were removed at the same time. Two evident peaks were not observed at an applied voltage of 70 mV/mm$^2$. However, two evident peaks were observed at applied voltages of 50 mV/mm$^2$ and 10 mV/mm$^2$ for SSCs with thicknesses of 2 mm and 0.5 mm, respectively.

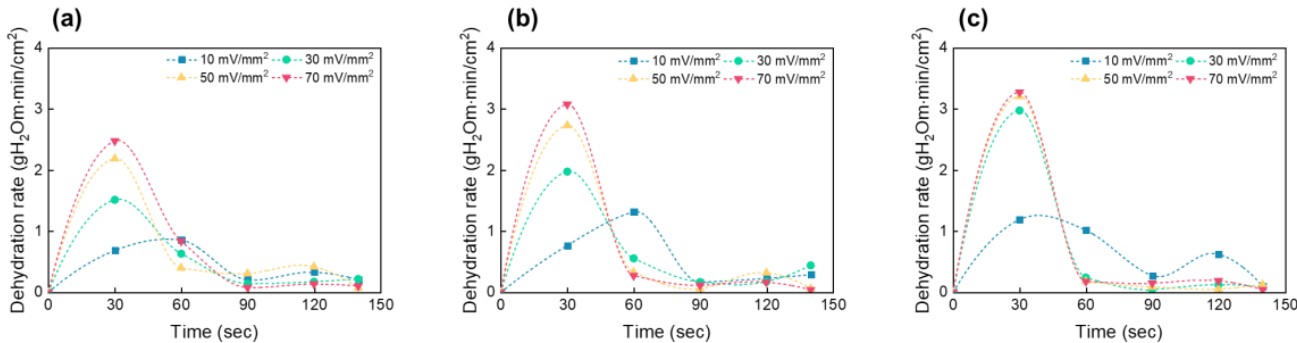

**Figure 3.** Dehydration rates of sewage sludge cakes with various voltages with respect to processing times of (**a**) 2 mm, (**b**) 1 mm, and (**c**) 0.5 mm.

Figure 4 shows the surface images of SSC after the ED process under operation duration of 140 s, SSC thickness of 1 mm, and at various applied voltages as follows: (a) as collected, (b) 10 mV/mm$^2$, (c) 30 mV/mm$^2$, (d) 50 mV/mm$^2$, and (e) 70 mV/mm$^2$. As mentioned earlier, the flocs interrupt the dehydration process as the bound water is stored inside. To obtain low moisture content and efficient dehydration, the flocs need to be destroyed, and the water inside the flocs should be removed. The flocs appeared to be uniform and intact up to an applied voltage of 30 mV/mm$^2$; however, above 50 mV/mm$^2$, the flocs appeared to be completely destroyed during the ED process. With the removal of bound water from SSCs, low moisture content and high dehydration rate are expected, which is in good agreement with the results shown in Figures 2 and 3. The moisture content should be <40 wt.%, for land dumping or to be used as SRF, and moisture content <50 wt.% can only be achieved when the flocs are destroyed.

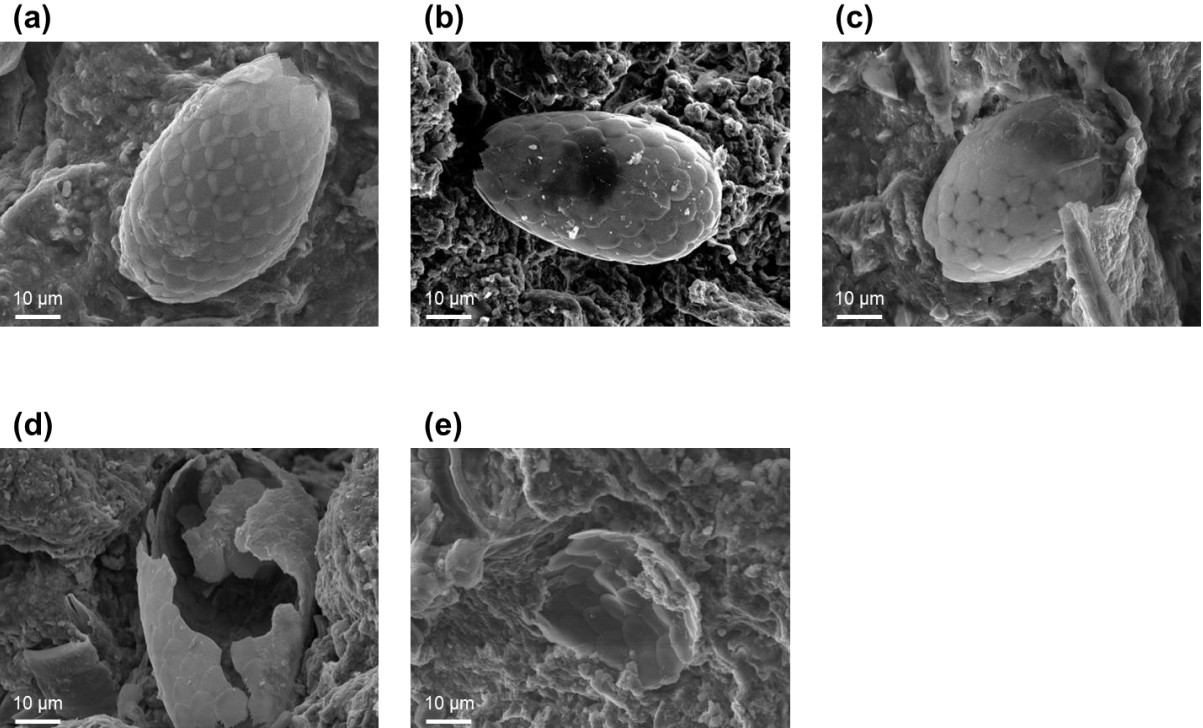

**Figure 4.** Floc images of sewage sludge cake after electrodehydration process at (**a**) as collected, (**b**) 10 mV/mm$^2$, (**c**) 30 mV/mm$^2$, (**d**) 50 mV/mm$^2$, and (**e**) 70 mV/mm$^2$ (ED processing time: 140 s, SSC thickness: 1 mm). Flocs in the sludge cake are clearly destroyed above 50 mV/mm$^2$.

Application of high voltage could wear out the anode and therefore decrease the lifetime of the IrO$_2$ anode. Therefore, the applied potential and morphological changes observed on the surface of the anode were used as the criteria for evaluating the lifetime of the anode. Figure 5a shows the cyclic voltammetry of the IrO$_2$ anode as-deposited and after the ED process of 1 mm thick SSC at applied voltages of 50 mV/mm$^2$ and 70 mV/mm$^2$ and at a processing duration of 140 s. A significant difference in the potential window was not observed (1.4 eV) [6,23]. It indicates that the lifetime of the IrO$_2$ anode was not affected under the applied potential of 50 mV/mm$^2$ and 70 mV/mm$^2$, which was initially assumed as a severe condition for the anode resulting in a decrease in their lifetimes. Figure 5b–d shows the surface morphology of IrO$_2$ anodes as follows: (b) as-deposited, (c) after ED at 50 mV/mm$^2$, and (d) after ED at 70 mV/mm$^2$. The morphology of the as-deposited IrO$_2$ anode shows typical mud-crack shapes on the surface. The surface morphology at 50 mV/mm$^2$ applied voltage does not show significant changes except the small extension of mud cracks due to the solution process. However, in the case of applied voltage of 70 mV/mm$^2$, partial delamination was observed. It is expected that the delamination only

occurs only on the surface region, and it does not affect the entire anode material. The potential window, which indicates the ability to undergo the electro-osmosis phenomenon, indicates that the ED process was stable.

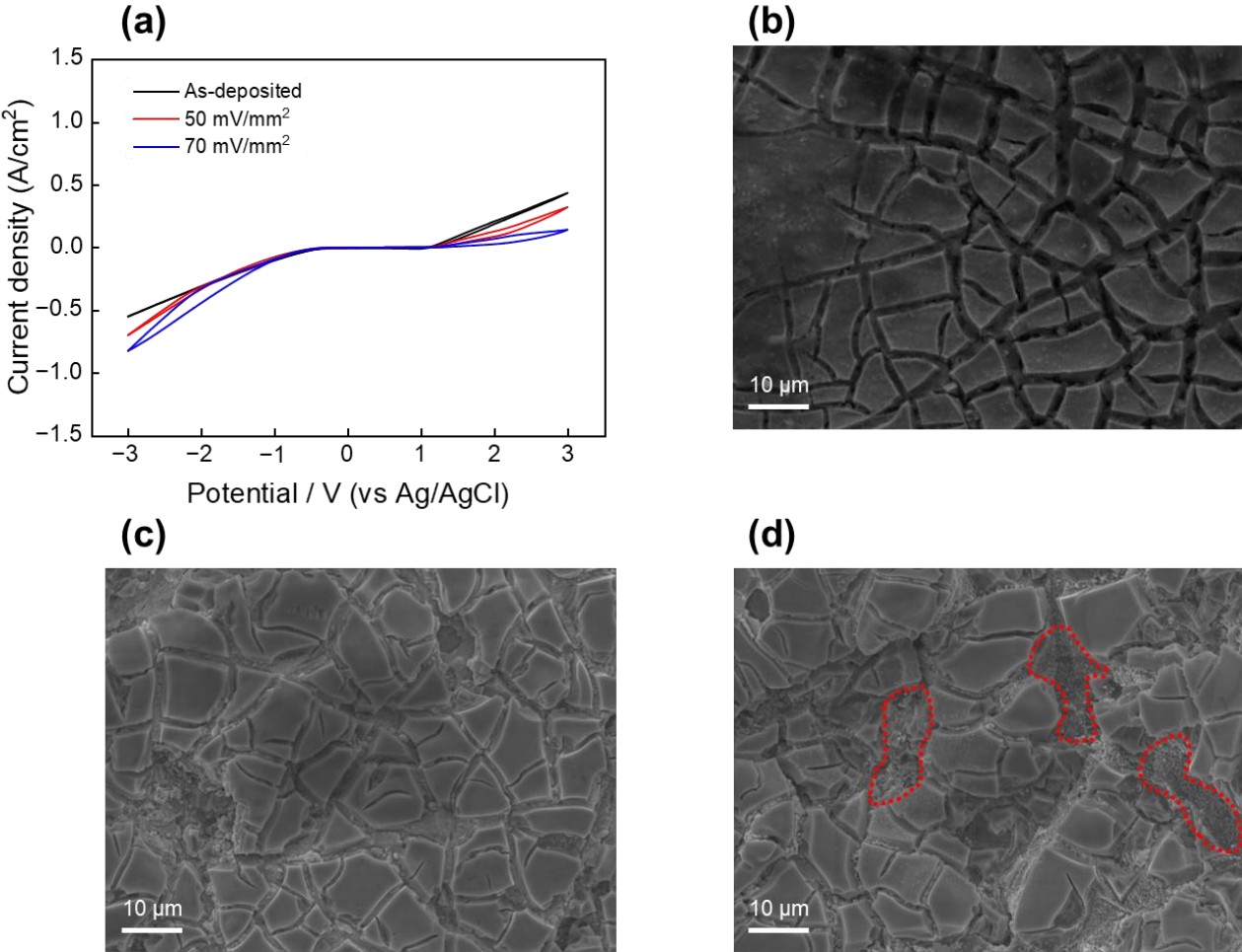

**Figure 5.** (**a**) Cyclic voltammetry curves and surface images (**b**) of as-deposited $IrO_2$, after ED process at (**c**) 50 and (**d**) 70 mV/mm$^2$. High-voltage processing does not show significant difference in potential window but causes delamination at 70 mV/mm$^2$ (ED processing time: 140 s, SSC thickness: 1 mm).

As surface morphology provides information only about the surface, and the XRD analysis can evaluate the entire anode due to deeper penetration of X-ray (~10 μm), XRD measurements were conducted to understand the lifetime of the anode after the ED process. Figure 6 shows the XRD patterns of the as-deposited $IrO_2$ anode and the anode after the ED process under the operating conditions of 1 mm SSC thickness, applied voltage of 50 mV/mm$^2$ and 70 mV/mm$^2$, and processing duration of 140 s. The red circle indicates the peak corresponding to Ti substrate, and the $IrO_2$ showed preferred orientation r (221) regardless of the operating conditions of the ED process. If the delamination of the surface has significantly influenced the microstructure of the $IrO_2$ anode and resulted in lattice contraction of extension, then the $IrO_2$ peaks should be shifted. But such a shift in the peak was not observed. Therefore, it can be concluded that the surface delamination observed at 70 mV/mm$^2$ voltage (Figure 5d) did not affect the lifetime of the anode. Significant evidence of the deterioration of the anode was not observed in the potential window and XRD evaluation of $IrO_2$.

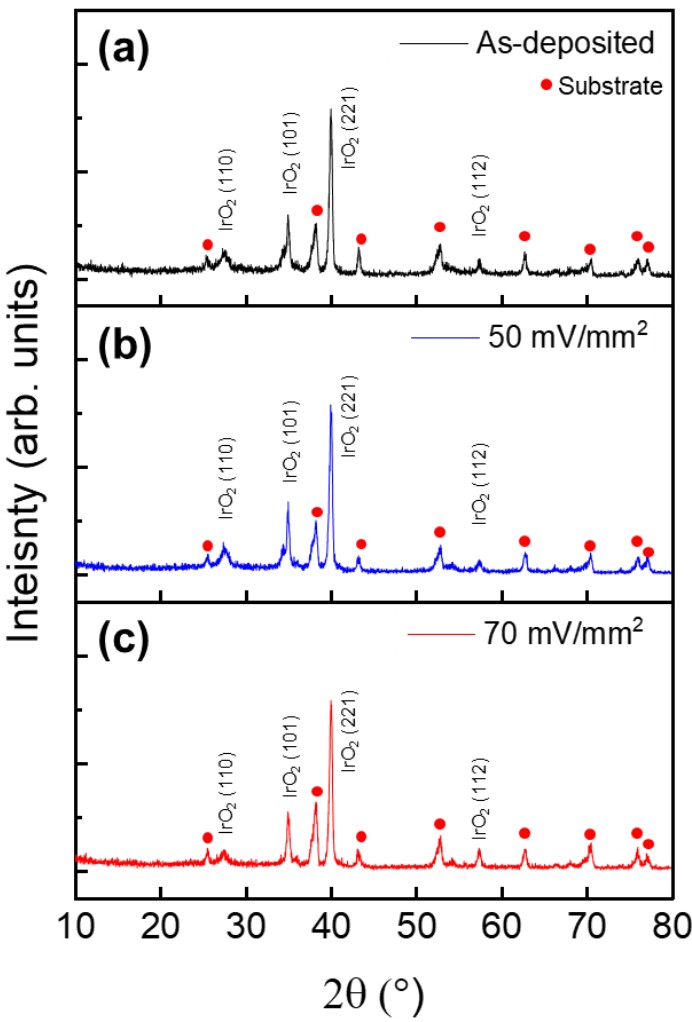

**Figure 6.** XRD patterns of (**a**) as-deposited, and after ED process at (**b**) 50 mV/mm$^2$ and (**c**) 70 mV/mm$^2$. Even 70 mV/mm$^2$ shows parts of delamination (Figure 5d), it does not show significant difference in XRD patterns (ED processing time: 140 s, SSC thickness: 1 mm).

## 4. Conclusions

IrO$_2$ anode material for the ED process was evaluated using a laboratory-made batch type ED system, and the operation conditions of the ED system were optimized. The key factors that mainly affect the ED process including applied voltage, processing time, and thickness of SSC were investigated. The moisture content proportionally decreased with decreases and increased with the thickness of SSC, and voltage and processing time. In the early stage of the ED process, dehydration from interstitial and surface water occurred, and then, the bound water that exists inside the flocs is removed. The proper dehydration that can be used as SRF (<40 wt.%) can be achieved only at applied 70 mV/mm$^2$ and at an SSC thickness of <1 mm. The dehydration rate at 50 and 70 mV/mm$^2$ clearly showed the removal of the interstitial and surface water, and also showed the removal of the bound water, which can be achieved via flocs destruction (Figure 5 d,e). The SSC of 0.5 mm thickness at 70 mV/mm$^2$ applied voltage showed the lowest moisture content and dehydration rate, but it is hard to adjust this for industry owing to its low dehydration capacity, which is directly connected to cost. Therefore, the dehydration system for SSC using IrO$_2$ as an anode can be summarized as 70 mV/mm$^2$ applied voltage, SSC thickness of 1 mm, and the processing duration of 140 s.

**Author Contributions:** Conceptualization, S.K. and M.-Y.Y.; methodology, S.K. and M.-Y.Y.; formal analysis, M.-Y.Y.; investigation, S.K. and M.-Y.Y.; data curation, S.K. and M.-Y.Y.; writing—original draft preparation, N.-Y.L. and S.K.; writing—review and editing, S.K., M.-Y.Y. and P.K.S.; visualization, S.K.; supervision, P.K.S.; funding acquisition, J.L. and P.K.S. All authors have read and agreed to the published version of the manuscript.

**Funding:** This work was partly supported by the Technology Development Program of MSS (S2780957) and by the Ministry of the Environment (G232019012551) and by the Korea Industrial Complex Corporation (HRBS2116), and partially supported by R&D Platform Establishment of Eco-Friendly Hydrogen Propulsion Ship Program (No. 20006644). This research was supported by Development and demonstration of on-board marine debris disposal modules program of Korea institute of Marine Science & Technology Promotion (KIMST) funded by the Ministry of Oceans and Fisheries (KIMST-20220494).

**Institutional Review Board Statement:** Not applicable.

**Informed Consent Statement:** Not applicable.

**Data Availability Statement:** Data sharing is not applicable to this article.

**Conflicts of Interest:** The authors declare no conflict of interest.

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
