# Peer review of "Performance of Insoluble IrO2 Anode for Sewage Sludge Cake Electrodehydration Application with Respect to Operation Conditions"

_coatings, doi:10.3390/coatings12060724_

Round 1

Reviewer 1 Report

This manuscript evaluates the performance of the sewage sludge cake electrodehydration (ED) system using IrO2 anode material. The operating conditions of the anode such as reaction time, sludge thickness, and voltage on SSC are investigated. The results may be referenced by the related investigations. The following points need to be addressed.

  1. What are the responding reactions in the process of electrodehydration (ED)?
  2. IrO2 is also expensive. May IrO2 be replaced by one cheap material (e.g., graphite)?
  3. Recent references are absent. Those recently published references need to be supplied and discussed.
  4. Conclusion section is a bit cumbersome. It should be concise one.

Author Response

Thank you for your kind comments and consideration for the publication in Coatings.

Please find attached files.

We look forward to receiving a response.

Reviewer 2 Report

The authors reported results on the performance evaluation of mechanical and electrochemical dehydration (ED) processes using IrO2 as an anode material for sewage sludge treatment to obtain sewage sludge cakes (SSCs). The authors show that using only the mechanical dehydration process, the water content of the SSCs is about 70-80% by weight, which is even higher and insufficient for landfilling or for use as a solid recycled fuel. The moisture content of the SSCs after treatment was used by the authors as a parameter to evaluate the performance of the ED process. They showed that this moisture content decreases in proportion to the reaction time, the thickness of the sludge, and the applied tension. Their results identified that moisture content of 40 wt.% is the optimum amount for landfilling the CSDs or for use as a recycled solid fuel.

The authors' results are of great interest to the readers of the Materials journal, and deserve to be published after considering the comments below:

  • The figures need to be improved, for example in SEM images 4 and 5 the scale bar is not clear, and in figures 2 and 3, the legends are not clear.
  • Ligne 191: the reference is missing.
  • Ligne 210: “was were” should be replaced by “were”, …… the manuscript needs linguistic revision.
  • Ligne 218: The authors state that coating delamination can be seen in Figure 5d, but it is a bit difficult to identify in the figure. The authors should illustrate it clearly in the figure and explain how it can be identified.
  • The authors should explain the driving force behind the formation of delamination, and they should explain why delamination will not reduce the life of the anode.
  • The authors should discuss the XRD results in depth by evaluating the stress and grain size information from the diffractograms.

Author Response

(The authors gave the same response as above.)

Round 2

Reviewer 1 Report

The previously raised points have been addressed or justified in the revised version, and the manuscript has been improved. The manuscript in the present form may be considered for the publication in this journal. 

Reviewer 2 Report

The authors have answered my questions well and taken my comments into account, so I recommend the publication of this article in its current form.